# Scientometric and Methodological Analysis of the Recent Literature on the Health-Related Effects of Tomato and Tomato Products

**DOI:** 10.3390/foods10081905

**Published:** 2021-08-17

**Authors:** Francesca Tilesi, Andrea Lombardi, Andrea Mazzucato

**Affiliations:** 1Department of Ecological and Biological Sciences, University of Tuscia, 01100 Viterbo, Italy; francesca.tilesi@unitus.it; 2Department of Agriculture and Forest Sciences, University of Tuscia, 01100 Viterbo, Italy; andrealombardi315@gmail.com

**Keywords:** antioxidant, anthocyanins, anticancer properties, carotenoids, food nutrition, phytochemicals, *Solanum lycopersicum* L., systematic review, tomato germplasm

## Abstract

The health benefits of tomato, a vegetable consumed daily in human diets, have received great attention in the scientific community, and a great deal of experiments have tested their utility against several diseases. Herein, we present a scientometric analysis of recent works aimed to estimate the biological effects of tomato, focusing on bibliographic metadata, type of testers, target systems, and methods of analysis. A remarkably variable array of strategies was reported, including testers obtained by standard and special tomatoes, and the use of in vitro and in vivo targets, both healthy and diseased. In vitro, 21 normal and 36 cancer human cell lines derived from 13 different organs were used. The highest cytotoxic effects were reported on cancer blood cells. In vivo, more experiments were carried out with murine than with human systems, addressing healthy individuals, as well as stressed and diseased patients. Multivariate analysis showed that publications in journals indexed in the agriculture category were associated with the use of fresh tomatoes; conversely, medicine and pharmacology journals were associated with the use of purified and formulate testers. Studies conducted in the United States of America preferentially adopted in vivo systems and formulates, combined with blood and tissue analysis. Researchers in Italy, China, India, and Great Britain mostly carried out in vitro research using fresh tomatoes. Gene expression and proteomic analyses were associated with China and India. The emerging scenario evidences the somewhat dichotomic approaches of plant geneticists and agronomists and that of cell biologists and medicine researchers. A higher integration between these two scientific communities would be desirable to foster the assessment of the benefits of tomatoes to human health.

## 1. Introduction

Vegetables are components of the human diet that are of crucial importance as sources of antioxidants, vitamins, and other essential health-related compounds. The tomato (*Solanum lycopersicum* L.) is the most widely used and versatile fruit vegetable, being consumed fresh (raw) or cooked or as a variety of processed products [1]. Worldwide, the tomato has reached an investment of over 5 million hectares and a production higher than 180 million tons; meanwhile, its average yearly per capita consumption reaches 17 kg [2]. Tomato is an important player of a balanced diet, due to its content of functional compounds, ranging from vitamins, minerals, dietary fiber, proteins, and essential amino acids to secondary metabolites, such as carotenoids, chlorophylls, and polyphenols [3]. A remarkable number of studies have addressed the effects of tomato bioactive components, such as purified molecules, extracts, or processed products, against a wide range of human diseases. This work has been exhaustively reviewed, both in papers that have considered the total content of health-related compounds [3,4] and in specific analyses, focused on carotenoids [5,6,7,8], flavonoids [9], or vitamins [10].

The genetic variability affecting tomato fruit pigments allows the diversification of fruit composition and, eventually, health-related properties. The tomato fruit is a berry that can assume different shapes and colors; standard tomatoes are red at ripening due to the prevalent accumulation of the carotenoid pigment lycopene, coupled with limited amounts of β-carotene. As dietary components, carotenoids contribute a high antioxidant potential and the ability to prevent the onset of certain cancers [5,6,8] and cardiovascular diseases [7]. In addition, β-carotene is the most potent dietary pro-vitamin A and, as such, it is required for normal cellular differentiation, development, immune function, and the transduction of light to neural signals [11]. Carotenoid synthesis is activated by phytoene synthase 1 (PSY1) [12]; *psy1* loss-of-function variants, known as *yellow flesh* (*r*) mutations, are the basis of the yellow fruit phenotype. In *r* fruits, phytoene and colored carotenoids are strongly reduced, whereas other carotenoids (xanthophylls) and polar compounds (amino acids, vitamins, and quinones) are increased [13]. Other carotenoid variants occur at the *lycopene β-cyclase* (*tlcy-b*) locus; the *Beta* (*B*) allele represents the wild-type form of the gene and entails an increase in β-carotene [14], whereas *old-gold crimson* (*og^c^*) is a severe loss-of-function allele that has been incorporated into many tomato varieties to increase the lycopene content [15]. Finally, the mutation *tangerine* (*t*) affects *carotenoid isomerase* (*CRTISO*); *t* mutants yield orange fruits that accumulate prolycopene instead of all-trans-lycopene, raising a nutritional interest for its higher bioavailability [16].

Ripe tomatoes do not normally contain chlorophyll, but fruits of *green-flesh* (*gf*) mutants retain chlorophyll and present higher levels of tocopherols (vitamin E) [13,17]. Because of the positive effects of chlorophylls and chlorophyll-related metabolites on cellular inflammation and as anti-mutagen and anti-carcinogen agents, *gf* mutants are considered biofortified tomatoes. When *gf* is combined with “high-pigment” gene variants, the fruit phenotype becomes so dark that these genotypes are often referred to as “black tomatoes” (e.g., Crimea black, Black prince, and Black plum) [9].

Flavonoids are secondary metabolites responsible for a large array of functions in the plant kingdom; they are also known for their positive effects on human health, including prevention of some types of cancer, cellular antioxidant activity, and hemolysis inhibition [18]. The occurrence of flavonoids in tomato fruits is restricted to the peel [19]; the *colorless fruit epidermis* (*y*) mutant shows defective flavonoid accumulation, giving fruit a pink appearance [20], with a decrease in naringenins and an increase in kaempferol and quercetins [13].

Although cultivated tomatoes do not normally produce anthocyanins in fruit, the combination of alleles from related wild species (*Anthocyanin fruit* (*Aft*) from *S. chilense* and *atroviolaceum* (*atv*) from *S. cheesmaniae*) has allowed purple-fruited genotypes to be bred [19,21]. Both genes belong to the MYB family of transcription factors [22,23]. The lines bred with this genetic combination have been registered with suggestive names such as “Sun Black” or “Indigo Rose” and are referred to as “purple” or “black” tomatoes. Anthocyanin-rich tomatoes have also been developed through genetic engineering, by driving the fruit-specific expression of regulatory genes from snapdragon and resulting in anthocyanin biosynthesis in all fruit tissues [24]. Breeding or engineering of high-anthocyanin tomatoes has raised great interest, as the antioxidant capacity in purple tomato fruits is higher than in standard tomatoes. The healthy properties of anthocyanin-enriched tomatoes, based on the capacity to act as scavengers of harmful reactive chemical species and inhibitors of proliferating cancer cells, has been demonstrated in both in vitro and in vivo studies [9,21].

A regular intake of vitamins represents a crucial aspect in human nutrition, because either low or excessive amounts of these compounds can contribute to the development of several diseases. Tomato represents an important source of vitamins, mainly of the water-soluble ones [3]. The highest levels are reported for L-ascorbic acid (AsA, vitamin C), essential for humans, who are unable to synthesize it. Variability for the AsA content in tomatoes has been reported [25], and genetic and environmental factors result in AsA variation across both cultivated varieties and wild species [26]. Vitamin E is also synthesized by photosynthetic organisms and consumed by animals within their diet; as mentioned, it increases in *r* mutants and in *r gf* combinations (green tomatoes) [13]. Epidemiological studies have reported that high vitamin E intakes are linked to a reduced risk of cardiovascular diseases, and that vitamin E shows synergistic properties with lycopene in inhibiting leukemia and prostate carcinoma cell proliferation [10].

Glycoalkaloids, such as solanine, a-tomatine, and dehydrotomatine, are mostly found in green tomatoes, where they play a role in plant protection against insects, fungi, and bacteria [27]. In the human diet, they are considered antinutritional components, but are also thought to have a variety of pharmacological and nutritional properties, including anticancer, antibiotic, anti-inflammatory, and antioxidant activities [28]. When fruit ripens, glycoalkaloids are largely catabolized; although their content in ripe cultivated tomatoes is low, genotypic differences exist [29].

In the present research, we aimed to analyze the recent literature dealing with the biological effects of tomato extracts estimated using in vitro and in vivo target systems. Whereas previous reviews largely addressed the inventory of obtained results [3,4,5,6,7,8,9,10], we focused on inventory scientometric details, the adoption and description of the tester material, the addressed target, and the level of analysis used. Our aim was to provide insights into the experimental systems of choice and the degree of integration of genetic, agronomic, biochemical, and medical aspects, and finally to assess the comparability of different experiments.

## 2. Materials and Methods

### 2.1. Search Strategy, Eligibility Criteria, and Data Extraction

This review considered 12 years of studies dealing with the beneficial effects of tomatoes on human health. The workflow to select contributions from the literature followed the recommendation of the PRISMA statement for reporting systematic reviews and meta-analyses [30].

The NCBI PubMed Database (https://pubmed.ncbi.nlm.nih.gov, accessed on 10 May 2021) was used to search articles published from 1 January 2008 to 30 April 2021. The key search terms were selected based on the efficiency of preliminary queries; “Tomato” was finally adopted in all queries, combined with “Anticancer”, “Apoptosis”, “Antioxidant AND Activity AND Human” and “Bioactive AND Compounds”. Reference lists from published studies were also manually searched to identify additional articles.

An initial screening of the retrieved literature was conducted based on pertinence, to remove studies that were not focused on the relationship between tomatoes and human health. Additional levels of screening included the exclusion of reviews, duplicates, and papers in journals in quartiles lower than Q2, according to the SCImago Journal and Country Rank (https://www.scimagojr.com/index.php, accessed on 20 May 2021), considering the year of publication and the best ranking category.

Full texts were collected and scientometric information recorded, including the country of provenience of all the authors, of the corresponding author, and the number of authors. Bibliographic information such as journal subject category, journal country, and journal quartile were also registered. Country names were reported using the ISO-3166 Alpha-3 codes (https://www.iso.org/iso-3166-country-codes.html, accessed on 1 June 2021). The 2019 impact factor (IF) was obtained through the Clarivate Analytics InCites website (http://journalprofile.clarivate.com/, accessed on 1 June 2021).

The tomato products adopted as testers were initially distinguished into fresh (F; fresh tomatoes to obtain extracts), purified (P; chemicals sold in purified or synthetic form), and formulate (FO; formulates for human assumption) forms. Processed tomatoes purchased from the market (pasta, passata, and juice) were classified as F testers.

For F products, it was recorded whether “standard” or “special” tomatoes were adopted, where special meant genotypes endorsed with quantitative and/or qualitative extra compounds. In addition, it was recorded whether details on the tomato genotypes and on the agronomic conditions adopted to produce them were reported. Details on the protocols used to obtain a powder, juice, passata, paste, extract (hydrophilic and lipophilic from the fruit, leaves, and seeds), or further fractionation or purification were also collected, as well as if a tester characterization was presented.

For P and FO testers, it was reported whether the authors detailed the supply sources and the adopted doses. FO testers included products obtained by procedures different from normal tomato processing, such as extraction, concentration, and formulation, to produce capsules, beadlets, pasta, or powder preparations. For FOs, it was also reported whether the authors detailed any characterization of the extract.

Regarding the target material, studies were distinguished if they adopted an in vitro or in vivo experimental system, or both. In in vitro studies, the type of cell lines used was recorded, with reference to the organism of origin (man, mouse, or other), to the organ/tissue of provenience, and to the sanitary status (healthy or diseased). In vivo experiments were classified according to the target organism (man, mouse, or other) and to its sanitary status (healthy, stressed, or diseased). The type of analyses performed was also recorded.

For studies addressing cytotoxicity in vitro using human cell cultures, the effect of the best tester on each used cell line was inferred and expressed as the percent gain of cytotoxicity in the treated versus the untreated thesis. This best performance (BP) value referred to sublethal doses in studies where lethal doses were reached in the control. In studies expressing cytotoxicity through the IC_50_ value, BP was calculated, when possible, by comparing the IC_50_ dose of the treated versus the untreated control. To include such estimation in the correspondence analysis procedures, the BP value was categorized into six classes (1 = BP ≤ 10%; 2 = 11–30%; 3 = 31–50%; 4 = 51–70%; 5 = 71–90%; 6 = BP > 90%).

All data were extracted from the published papers using a systematic procedure carried out by at least two review authors. Every effort was made to adopt equal estimation parameters in a bona fide way and ensure the right interpretation of the information.

### 2.2. Statistical Analysis

Differences among the BP values in the different organs or cell lines were estimated using the PROC GLM implemented in the SAS software package (v9.4M6, SAS^®^ University Edition, SAS Institute Inc., Cary, NC, USA) after arcsine transformation to meet variance homoscedasticity. Multiple correspondence analysis was carried out using the PROC CORRESP procedure implemented in SAS and the results were plotted according to the first two dimensions. Variable levels in the correspondence analysis for the country of the corresponding author and the analysis type were limited to the most populated five and nine classes, respectively. For the tester type, it was specified if the F products were derived from standard or special tomatoes. When the target type was related to the ester type, it was specified if the target was a human or a murine system.

## 3. Results

### 3.1. Geographic Distribution of Authors and Journal Category

A total of 803 titles were retrieved in the database search; approximately 68%, 10%, and 5% of them were discarded as non-pertinent, reviews, or duplicates, respectively (Figure 1). Finally, 26 publications were excluded as they did not meet the quartile criterion, and the final total number of selected papers was 107 (Figure 1 and Appendix A) [31,32,33,34,35,36,37,38,39,40,41,42,43,44,45,46,47,48,49,50,51,52,53,54,55,56,57,58,59,60,61,62,63,64,65,66,67,68,69,70,71,72,73,74,75,76,77,78,79,80,81,82,83,84,85,86,87,88,89,90,91,92,93,94,95,96,97,98,99,100,101,102,103,104,105,106,107,108,109,110,111,112,113,114,115,116,117,118,119,120,121,122,123,124,125,126,127,128,129,130,131,132,133,134,135,136,137].

Authors from 34 countries were involved in the selected studies, with Italy (ITA) and the United States of America (USA) presenting the highest occurrences (Table 1). Such countries remained predominant also when referred only to corresponding authors; overall, papers with a corresponding author from ITA or USA showed the highest mean number of authors (Table 1). Among countries with fewer authorships, The Netherlands (NDL) and Switzerland (CHE) had a single contribution, but a high number of papers published in their journals (24 and 15, respectively; Appendix A). Germany (DEU) and Japan (JPN) were also among the countries with less than three papers. Eastern Asian countries, such as the Republic of Korea (KOR), Malaysia (MYS), and China (CHN), showed the highest degree of international collaboration; by contrast, none of the nine papers published by authors from India (IND) were shared in international collaboration (Table 1). More than 50% of the papers were published in journals issued in the USA or the United Kingdom (GBR), whereas countries with a high number of studies, such as ITA and IND, did not present any favored journal (Table 1).

Out of the 107 papers reviewed, 71 were in Q1 journals at the time of publication and 36 in Q2. The most represented journal categories were “medicine” and “biochemistry, genetics, and molecular biology”, with a slightly higher distribution in Q1 (Figure 2). The term “agriculture” ranked third, with an absolute number of occurrences in Q1 journals higher than in “medicine”. An important category such as “nursing” that includes the area “nutrition and dietetics” was found only for 27 journals, followed by “chemistry” and “pharmacology, toxicology, and pharmaceutics” (Figure 2 and Appendix A). “Pharmacology” was the only term where the occurrences in Q2 were higher than those in Q1.

To further characterize the favored journals, the journal IF was reported; seven papers were in the top class (IF > 6.0), nine in the second (5.9 > IF > 5.0), 29 in the third (4.9 > IF > 4.0), 25 in the fourth (3.9 > IF > 3.0), 26 in the fifth (2.9 > IF > 2.0), and 11 in the last (IF < 1.9) (Appendix A).

### 3.2. Materials Used as Tester: Fresh, Purified and Formulate Products

A first distinction was made among the use of testers from fresh (F), formulate (FO), or purified (P) products. Sixty-three studies adopted F tomatoes (Table 2); 26 provided no information on the tomato used, 15 reported the use of only standard tomato, and 22 reported the adoption of one or more special types (Appendix A). Although most of these studies used extracts from ripe fruits (15 used processed products that were either commercial or prepared ad hoc), one study adopted seed extracts, three vegetative tissues, and one fruits in immature stages (Table 2). Tomato seed extracts showed anti-proliferative and radical scavenging properties with interesting genotype-dependent results [78], whereas oilseed, used in an FO preparation in a study based on human macrophages, induced an increase in antioxidant power when compared to lycopene [69].

In studies that used vegetative parts, ref. [101] tested foliar extracts from different genotypes against a human gastric cancer cell line and reported cytotoxic effects in terms of radical scavenging activity, while [87] tested the antioxidant and antibiotics properties of phenolic and glycoalkaloid foliar extracts from two tomato genotypes with significant results against pathogenic bacteria.

Of the studies with F testers, six referred to the use of tomato waste [59,61,78,90,123,133]. Overall, approximately 87% of studies adopting F tomatoes reported a biochemical characterization of the extract.

Seventeen studies adopted FOs, three of them in association with F products (Table 2). Most studies using FOs reported the composition and concentration of the active compounds in the formulates, and approximately half of them measured such levels in the final diet, generally with reference to carotenoids; two studies also reported vitamin levels [69,118] (Appendix A).

Among the 38 studies that adopted P compounds, the majority investigated the effects of lycopene and tomatine (Table 2). In all studies except one, the source of the chemical was reported, whereas the adopted doses were always notified (Appendix A).

The type of processing, extract preparation, and fraction/molecule addressed were extremely variable; most studies referred to the hydrophilic or lipophilic extract or both (Table 2). Twenty-one out of the 26 studies that provided no information on the tomato source analyzed the extract, generally with reference to lycopene/carotenoids, or reported its analysis from previous work. Of the studies that adopted standard tomatoes, 87% reported an extract analysis, including carotenoids and other compounds (mainly phenols); such a ratio was 91% for the studies using special varieties (Appendix A).

None of the studies that lacked information on the tomato adopted added any insight on the agronomic conditions used. Reference to the tomato genotype was reported in almost all papers referring to standard and special tomatoes; still, a relevant number of papers lacked information on agronomic conditions (Appendix A).

### 3.3. Use of “Special” Tomato Genotypes as a Tester

Out of the 63 studies that used F material, 22 adopted one or more “special” tomato types, either including a normal counterpart or not (Table 3 and Appendix A).

Wild species were assessed in a single study, where their higher content of biologically active compounds was shown [91]. In [75], the authors adopted introgression lines with only specific genome portions from a wild species that harbored genes enhancing the AsA and phenolic content; thus, the positive effect of this enrichment was demonstrated in a cultivated genetic background.

Moreover, Petruk and co-workers assayed a high-AsA breeding line, demonstrating the ability of its extract to counteract UVA-oxidative stress on human keratinocytes [95]. This protective effect was due to the high concentration of vitamin C that acts as a free radical scavenger. 

Several papers addressed the peculiarity of landraces, such as varieties from the Campania region in Italy [78,98,130,133], whose extracts were active against an in vitro model of gastric cancer cell lines, without toxic effects on non-tumoral cells. In selections of landraces from Argentina [91] and Romania [123], a wide genotypic variability was observed for all of the parameters measured and, in both cases, landraces that stood out over commercial genotypes were evidenced.

Several authors used carotenoid mutants, either yellow- (*r* mutants; [67,71,78,89,91]) or orange-fruited. The latter represented *t* [71,85,124] or supposed *B* variants [67,71,124]. Moreover, a transgenic, orange-fruited genotype, obtained by overexpression of the tomato *tlcy-b* cDNA, was adopted [39]; HT-29 human colon adenocarcinoma cells treated with the extract from these fruits showed dose-dependent cell growth inhibition that was enhanced by the heat treatment of samples before digestion. Finally, one study adopted mutants enriched with lycopene due to the *og^c^* mutation [117], thus testing a variant that is already largely present in commercial tomato cultivars.

Although not always explicitly recognized in studies, five used simple *gf* mutants [67,71,78,127,130], one a *gf gs* combination [93], and one a couple of green-fruited genotypes that were supposedly due to the combination of *gf* and *r* [71].

A single study took into consideration the use of pink tomatoes, showing that these variants do not excel in carotenoid content nor in biological activity, ranking always least among lycopene-containing tomatoes [67].

Five studies carried out with variants affecting pigment composition included high-anthocyanin tomatoes; these works were generally carried out with the comparison of a red-fruited control and evidenced a superior bioactivity of anthocyanin-enriched tomatoes [67,68,77,111,116].

Finally, a study tested mutants obtained by space mutation breeding that presented variation in plant height and productivity or shelf-life and disease-resistance [49]. The mutant lines showed higher carotenoid, flavonoid, and vitamin C content with respect to their near-isogenic control.

### 3.4. Target Material

A first classification based on the target material indicated that most studies (*n* = 56) adopted in vitro experiments, 39 had an in vivo experimental system, and 12 had both (Appendix A).

Of the studies adopting in vitro tests, 59 were carried out with a human and 10 with a murine target; a minority of studies adopted other organisms (Table 4). Twelve and 35 studies used human normal and cancer cell lines, respectively; 18 adopted both targets, including normal lines as a control (Table 4 and Appendix A). The most used control cell lines were from the blood and lungs, whereas the most adopted cancer systems were from the colon, cervix, prostate, and breasts. In total, 21 different normal and 36 cancer human cell lines were used; the most common among the latter were HeLa and HT29 (Table 4).

The experiments that adopted murine systems mainly used normal cells lines challenged with inflammatory and oxidative stresses; three studies used cancer lines (Table 4).

Out of the 51 studies that adopted in vivo experimental systems, 18 were carried out on humans, 31 on mice, and one each on zebrafish (*Danio rerio*) and drosophila (*Drosophila melanogaster*; Table 5). In humans, the studies addressing tumoral patients were carried out only on men affected by prostatic cancer, whereas the others adopted healthy or stressed patients. Three studies with healthy individuals specifically addressed sportsmen (Table 5).

More in vivo research on tumoral illness was carried out with mice, including several targets such as blood, colon, liver, tongue, and prostate cancer (Table 5). In addition, a study investigated the effects of a fruit-specific cystine-knot miniprotein of tomatoes on in vitro endothelial cell migration and in vivo angiogenesis using a zebrafish model [92], while [113] adopted, together with an in vitro human leukemia cell system, an in vivo model with drosophila to evaluate the biological activity of tomatoes and lycopene as a chemopreventive agent. Interestingly, these authors also focused on estimating tomato and lycopene “cytotoxicity,” showing that they do not induce internucleosomal DNA fragmentation. 

Eight in vivo studies that adopted live mice, as well as zebrafish and drosophila, addressed in parallel human cell lines, thus testing the same effector in two different species. Study [48] was the only one that used human targets both in vitro and in vivo, focusing on the absorption of lycopene related to its isomerization state (Appendix A).

### 3.5. Type of Analysis and Anti-Proliferative Effects on Human In Vitro Cell Cultures

The main analytical aspects taken into consideration by the 107 studies were categorized into 11 classes; “cell viability and cytotoxic effect” was the most investigated parameter, followed by “antioxidant capacity,” with 52 and 48 studies, respectively (Figure 3). The third most populated analysis class was “proteomics and immunology”, whereas “tissue analysis” and “blood analysis,” two approaches typical of in vivo studies, came fourth and fifth. Other levels of study, such as “apoptosis,” “gene expression,” and “cell cycle” were prevalent in in vitro trials; “inflammatory activity” was equally distributed between in vitro and in vivo studies (Figure 3).

A summary of the main effects obtained in each reviewed study is reported in Appendix A. Here, the cytotoxic and anti-proliferative properties of tomato extracts on human in vitro cell cultures was evaluated by inferring the effect of the best extract in each study expressed as a gain in toxicity of the treated versus the untreated thesis (BP). A BP value could be inferred in 89 events related to 18 target organs, with a minimum of three for kidney (only organs with >2 data points were included in the analysis) and a maximum of 21 for colon (Appendix A). Although no significant difference was reported among organs, “breast” scored the highest mean BP value (Figure 4a). Overall, the effects on normal cell lines were investigated only 12 times and related to nine targets (Appendix A). In all organs, the effect on cancer cell lines was higher than that reported in normal controls, although a statistical analysis was not possible due to the low number of studies that adopted control lines (Figure 4a).

Considering single cell lines (only lines with >2 data points), the highest BPs were found for LNCaP and MCF-7, which represent a prostate and breast targets, respectively (Figure 4b). Also in this case, statistical differences among lines were not found.

### 3.6. Multivariate Analysis

Correspondence analysis involving the variables “country” (of the corresponding author), “tester,” and “target”, indicated that researchers from the USA preferentially adopted in vivo target systems and FO testers, whereas in vitro research using F tomatoes was carried out mostly in ITA, IND, and ESP (Figure 5a). The use of special tomatoes was mainly associated with ITA, as was the use of both target types and P testers (Figure 5a).

When the variable “country” was correlated with the “analysis type,” the USA were associated with the use of blood and tissue analysis, whereas ITA and ESP used the detection of antioxidant and anti-inflammatory capacity more (Figure 5b). CHN, together with IND, resulted associated with gene expression and proteomics (Figure 5b).

Considering “journal category,” agriculture was associated with the use of F tomatoes, both standard and special; by contrast, the categories medicine and pharmacology were associated with the use of FO testers (Figure 5c).

Research with cell lines was generally carried out with F and P testers (Figure 5d). The adoption of F testers was prominent to test the effect on in vitro systems, both human and murine, whereas in vivo research mostly used FOs (Figure 5d).

The study of cytotoxicity was related mainly to the use of human in vitro systems, whereas in vivo experiments were associated with tissue and blood analysis (Figure 5e). Finally, a higher BP was more frequently obtained with P testers (Figure 5f).

## 4. Discussion

### 4.1. Bibliometrics Trends

Although authors from 34 different countries contributed to the selected literature, the primacy of a few countries was evident. ITA and the USA emerged for their number of authorships; however, whereas the USA also stood out for the number of journals involved, ITA did not present any journal of choice. The prominence of ITA, the USA, and ESP in the number of papers was positively related to the mean tomato consumption in those countries (Appendix A). Eastern Asian countries, such as CHN and KOR, were fertile in producing papers on the subject, with the highest degree of international collaboration. Considering journal categories, most papers belonged to the biomedical area (“medicine” and “biochemistry, genetics, and molecular biology”); papers published in “agriculture” journals ranked third, but with the highest proportion of Q1 journals. Although papers were selected in the Q1 and Q2 quartiles, the journals spanned a range of IF from 1.10 to 7.18, indicating that the selected literature covered a wide variety of journals.

### 4.2. Use of Different Testers: Fresh, Formulate, and Purified Products

A wide variety of testers was adopted in the reviewed studies, including F, FO, and P products. The adoption of F material allowed to estimate possible synergistic effects of the tester [49], to select special tomatoes, and to assess a pool of compounds when semi-specific extracts were used. By contrast, although the administration of F testers is related more to tomato consumption as part of the diet, they may compromise result reproducibility and interpretation, as genotypic and agro-ecological factors impact metabolite composition. The biosynthesis of secondary metabolites is susceptible to growth conditions, abiotic and biotic stresses, and cultivation practices, but only in few cases were these variables considered in the reviewed articles. Ripening is also a crucial phase in determining the content and availability of biomolecules; only two studies used immature tomatoes as testers [44,97]; the higher bioactive effect of unripe tomatoes has been related with glycoalkaloid content, thus suggesting the opportunity for selecting new tomato varieties with higher tomatine content at maturity [44].

Using processed products, a strategy adopted in 15 studies, represented a realistic choice, as paste, ketchup, or gazpacho are foods consumed in the diet; in addition, processing conditions are critical events that affect bioavailability, with possible positive and negative effects [38,45,61]. The effect of thermal treatment was investigated in two articles [89,93], indicating a reduction in phenols, an increase in flavonoids, and stability of lycopene and total carotenoids. The interaction between single food components plays a crucial role in understanding the “food synergy” concept, which should be taken into consideration, together with processing, seasoning, and cooking, when estimating the beneficial effects of foods [132]. Conversely, the whole food approach cannot associate the biological activity detected for any compound or family of phytochemicals present in the tester food [119]. Finally, following the principles of the circular economy, some studies investigated the potential interest of processing wastes that could be recycled with the extraction of useful bioactive compounds to produce nutraceutical formulations [90,133,138]. Studies on vegetative tissues or on seed extracts showed the potential of tissues different from the fruit itself as sources of useful low-cost nutraceutical resources.

By contrast with F testers, P products offer the advantage of analytical grade chemicals that guarantee high reproducibility and precise dosage. P products are, however, not subjected to crucial processes such as in vivo biosynthesis and post-translational protein modification and preclude testing synergistic effects. The adoption of FOs represents an intermediate choice, offering a tester more complex than P and still more standardized and controlled than F products. FO testers were revealed to be useful for trials involving humans for their ease of supply. A drawback of FOs is that, usually, information on the source product was not available.

A high proportion (92%) of papers using F and FO testers presented extract characterization, carried out by chromatography or HPLC. Spectrophotometric quantification was also adopted in few works, offering a fast, but less accurate, assessment of several compounds or groups of compounds at a relatively low cost.

### 4.3. The Use of “Special” Tomatoes

The 22 studies that adopted this strategy addressed the properties of fruits that presented enhanced, or diversified, levels of carotenoids, flavonoids, anthocyanins, vitamins, and chlorophylls. In all, details on the genotypes used were provided; knowledge of the tomato type (the involved mutation, as well as the genetic background) is crucial information, as the genotype represents one of the most influential variables in terms of different responses.

In many cases, the effects of special tomatoes were studied without the comparison of red-fruited counterparts, thus obscuring the assessment if the specialty effectively involved an added value in biological activity. When a red reference was used, it was usually not a near-isogenic material, a condition that would ideally permit the best identification of effects peculiar to the genetic variant. However, developing near-isogenic lines is a long and cumbersome process, and a true near-isogenicity is difficult to obtain. Narrow-sense near-isogenicity can only be reached in transgenic materials (e.g., [24,39]), but transgenic varieties are not permitted in several countries and still experience consumer diffidence. In this scenario, new perspectives to develop near-isogenic variants are opened by genome editing protocols that are very effective in recapitulating monogenically controlled phenotypes and offer experimental systems more acceptable for trials with humans [139].

The selection of anthocyanin-rich tomatoes represents a novelty in the panorama of special testers [9]. In five works, the authors tested in vitro the anti-proliferative [68,77,116] and antioxidant [67,111] properties of these materials. One paper described a trial carried out with in vivo targets [77]. The advantages of anthocyanins, based on their thermal stability and anticancer properties, were described for several fruit crops such as blueberries [140], grapes [141], and pomegranates [142]. In works using “black” or “purple” tomatoes, the origin of the dark color was sometimes inappropriately misinterpreted, which could be due to the accumulation of chlorophylls (*gf* mutants) or anthocyanins (*Aft/atv* combination). For instance, Li and co-workers [67] referred to both V118 and Cherokee Purple as purple tomatoes, but the first is an anthocyanin-rich [143] and the second a *gf* type [17].

### 4.4. The Target Systems: In Vitro versus In Vivo Experiments and Analyses

In vitro experimental designs offer several advantages, based on limited costs, major reproducibility, minimized biohazard, and low ethical issues. As many bioactive tomato compounds exert their beneficial effect as radical scavengers or antioxidant agents, the most evaluated analysis category was represented by “antioxidant capacity”, being considered in approximately half of the studies.

Several cell lines were employed to test the anticancer properties of tomatoes, but only in a few cases in association with normal lines as controls. Cell viability was evaluated in 65% of articles that used an in vitro system, showing a dose-dependent effect of different extracts or of single tester molecules. The cytotoxicity of testers toward normal cell lines, when assessed, generally showed lower values; however, this aspect was seldom taken into consideration and deserves more attention in future research.

The wide variability of approaches, testers, targets, and experimental protocols makes it impossible to compare the results reported in the selected literature. An estimation on cytotoxic effects exerted toward human in vitro targets indicated mean BP values ranging between 50% and 70% when cancer cell lines were pooled within organs, and between 40% and 70% when single most-used lines were considered. Although the variability among experiments obscured statistical differences among organs or lines, it was clear that normal cell lines, when included in the experiments, were less sensitive to tester effects. Thus, in general, treatments showed selectivity toward cancer cells.

The use of cytofluorimetric protocols allowed cell cycle analysis to be investigated in 11% of the studies, whereas apoptosis, which can be evaluated through different reliable techniques [144], was addressed in more than 31%. Necrosis analysis was only cited in six articles.

Gene expression studies represent one of the most versatile methodologies to understand the real-time mechanisms of biological responses, which was carried out in approximately 27% of papers, whereas 40% addressed “proteomics and immunological analysis”.

In vivo studies allow the effect of bioactive compounds to be addressed against physiologically stressed patients, including subjects challenged by diabetes, asthma, hypertension, and obesity. The effect of lycopene in reducing bodyweight gain and fat mass content has been reported in different studies, with both human and murine targets [88,108,135], showing that the dietary intake of tomato can counteract this physiological disorder.

In vivo studies often performed blood analysis and tissue evaluation, but, if we consider the results of trials with humans, only in a few cases were in vitro trends confirmed. One of the most challenging aspects of testing extracts in vivo is represented by estimation of in-body bioavailability. Digestion is considered the critical aspect of bioavailability, because of the aggressive action of stomach acids. The effects of in vitro digestion were evaluated in two articles; author [39] tested a digestate from genetically modified β-carotene-rich tomatoes on colon cancer cell lines, obtaining a dose-dependent effect on cell growth, cell cycle, induction of apoptosis, and downregulation of cell cycle-related genes. In [110], digested and undigested tomatine provided similar results and influenced several genes related to cell cycle, apoptosis, tight junctions, sterol biosynthesis, and glucose and amino acid uptake.

### 4.5. Multivariate Correspondence Analysis

Non-parametric multivariate analysis indicated geographical trends, as the USA research works preferentially adopted in vivo systems and FO as testers, whereas others (e.g., ITA, IND, and ESP) mostly carried out in vitro research using F tomatoes. The use of special tomatoes was prevalent in ITA, reflecting the importance covered by this vegetable in the Mediterranean diet and the accessibility to researchers of landraces, heirlooms, and genetic variants. The prevalence of gene expression analyses was associated with studies carried out in CHN and IND. Journals indexed in the agriculture category were associated with the use of F tomatoes; by contrast, medicine, biochemistry, pharmacology, and nursing journals were more associated with the use of P testers and FOs. This reflects the dualistic approach of these kinds of study, which may be set up by plant geneticists and agronomists, that address the properties of F products and publish in agriculture-related journals, or by biochemists and medical researchers, who are more focused on the target, use P or FO testers, and publish in the biomedical literature.

In addition, multivariate analysis evidenced the relationship between studies conducted in vitro with the use of F and P testers and the investigation of cytotoxicity and apoptosis; in contrast, in vivo studies more frequently adopted FO testers and tissue and blood analyses. Finally, the strongest effects in terms of cytotoxicity were detected with the use of P testers, indicating that doses supplied with testers more calibrated on dietary assumption may reveal a generally lower bioactivity.

## 5. Conclusions

The present analysis of the literature on the biological effects of tomato highlighted a wide variability of all experimental design facets. Most of the studies presented promising results, although their comparability was hampered by the lack of standard procedures and guidelines. Several issues of potential interest for future research emerged, such as the importance of the ripening stage, the specialty of genetic variants, the effects of digestion, and the correlation between in vitro and in vivo experiments. Fostering these topics, plant geneticists could produce novel bio-fortified tomato varieties, with higher health benefits and resilience of bioactive compounds against extraction, digestion, and processing.

The present inventory gives an indication that the best approach to assess the health benefits of tomato and tomato products can only be reached with the contribution of all the players of this scientific chain, that starts with plant selection and ends with clinical trials. As the expectation on the impact of the diet and of fortified nutrients on human health is high and citizens show a growing interest on nutraceutical aspects of food, the establishment of specific cluster and cooperation programs is advisable.

## Figures and Tables

**Figure 1 foods-10-01905-f001:**
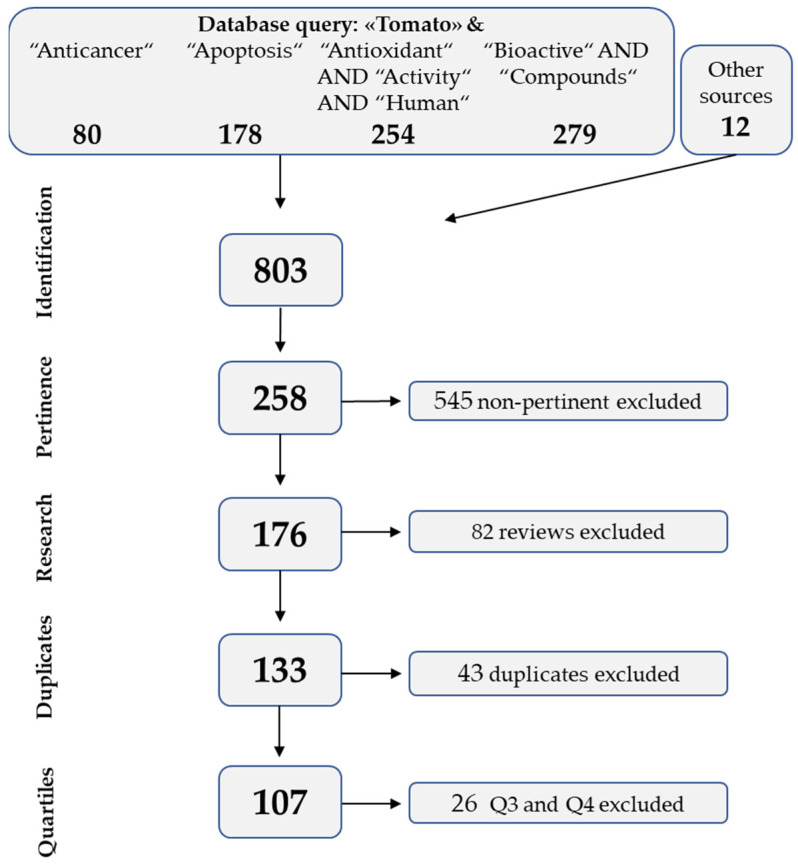
Flux diagram of the selection process and number of publications retained (in bold) or excluded at each step.

**Figure 2 foods-10-01905-f002:**
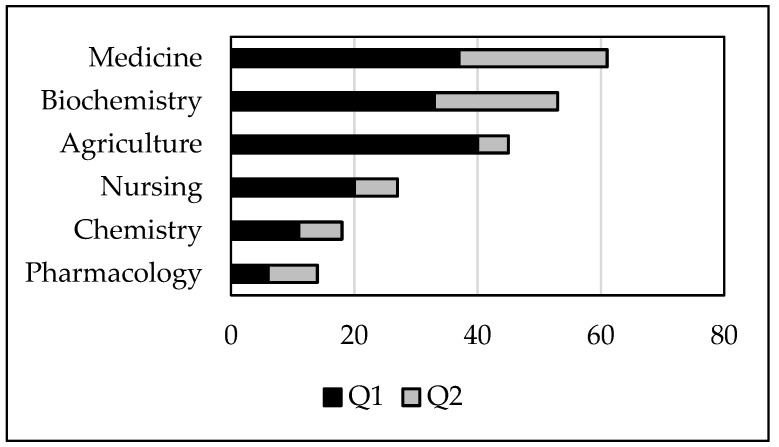
Distribution of the selected papers according to the journal subject category and quartile ranking.

**Figure 3 foods-10-01905-f003:**
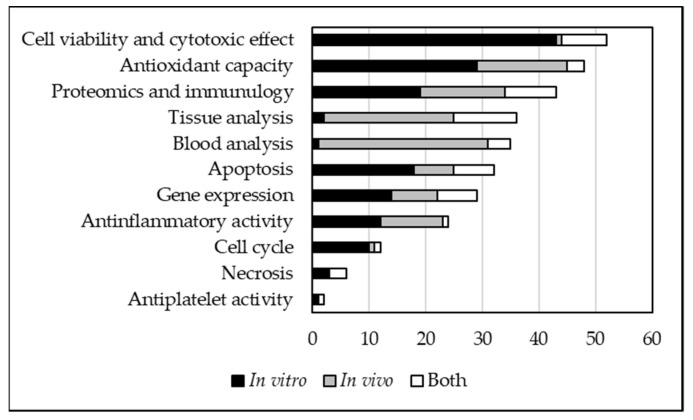
Type of analysis carried out in relation to the use of in vitro, in vivo, or both types of experimental system.

**Figure 4 foods-10-01905-f004:**
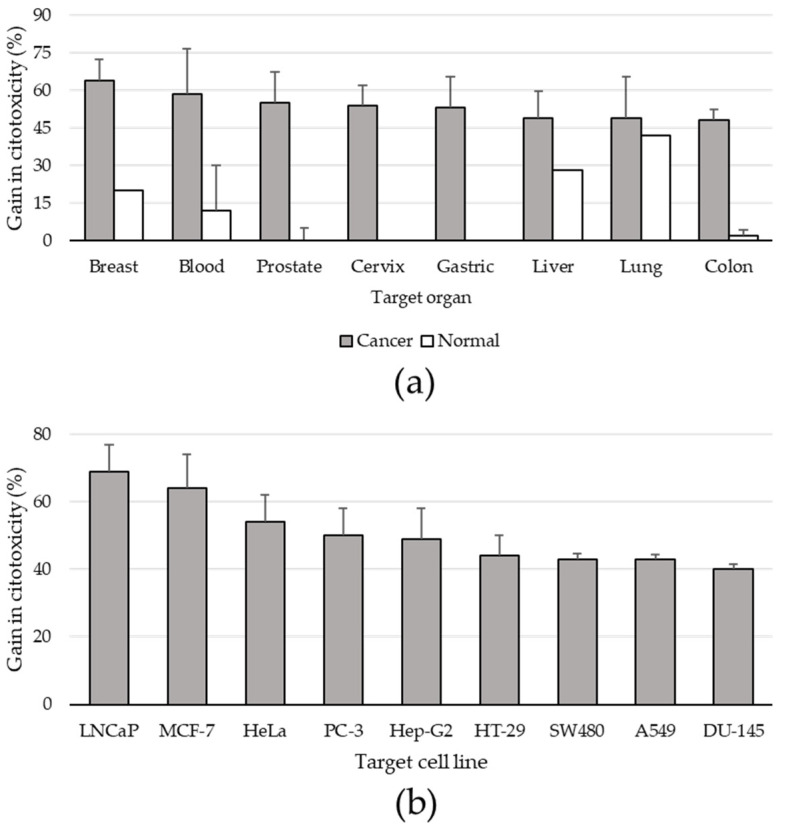
Percent cytotoxic effect of the best extract in different cell lines grouped for target organ (**a**) and as single cell line (**b**). Only targets with at least three inferred values were analyzed and reported; bars represent standard error of the mean.

**Figure 5 foods-10-01905-f005:**
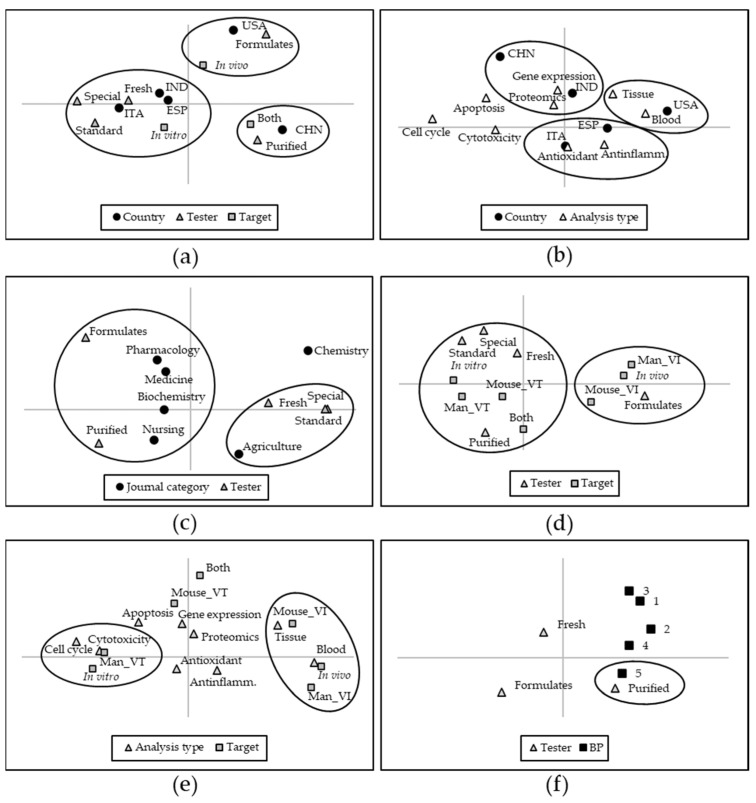
Correspondence analysis of the selected variables recorded in the text. (**a**) Variance of the “country” of the corresponding author (five most frequent countries) related to the “tester” (F, fresh tomatoes; FO, formulates; P, purified compounds) and “target type” (in vitro, in vivo, and both targets). (**b**) Variance of the “country” (as in (**a**)) related to “analysis type”. (**c**) Variance of “journal category” related to the “tester” (as in (**a**)). (**d**) Variance of the “tester” (as in (**a**)) related to the “target” (in vitro or in vivo experiments on man or mouse). (**e**) Variance of the “analysis type” as related to the “target”. (**f**) Variance of the “tester” in relation to the best performance (BP) index (for details, see the Materials and Methods section). Black circles highlight the associations of interest commented on in the text.

**Table 1 foods-10-01905-t001:** Scientometric data of 107 publications dealing with the nutraceutical assessment of tomato extracts and metabolites.

Authors’ Country	Country Code ^1^	No. of Papers			Mean No. of Authors	Journal Country
		Total	With Corresponding Author	With International Collaboration (%)		
Italy	ITA	22	22	22.7	9.0	0
United States of America	USA	18	12	38.9	7.9	37
Spain	ESP	13	11	15.4	6.2	1
India	IND	9	9	0.0	4.3	0
China	CHN	8	6	50.0	7.5	2
Australia	AUS	6	3	50.0	4.3	0
Portugal	PRT	5	4	40.0	4.6	0
Brazil	BRA	4	4	25.0	6.8	0
Republic of Korea	KOR	4	3	75.0	5.8	1
Chile	CHL	3	3	0.0	5.7	0
Great Britain	GBR	3	3	33.0	4.0	23
Malaysia	MYS	3	3	66.7	4.3	0
Taiwan	TWN	3	3	0.0	4.7	0
Other ^2^		27	21	37.0	4.9	43
**Total**			**107**		**6.3**	**107**

^1^ ISO-3166 Alpha-3 code. ^2^ Includes all countries with less than three publications: ARG, CAN, CHE, CMR, CYP, CZE, EGY, DEU, IRN, ISR, JPN, MEX, NDL, NZL, POL, ROU, SRB, SWE, and TUR with at least one corresponding author, and DNK and NOR with no corresponding author.

**Table 2 foods-10-01905-t002:** Type of testers used in the 107 selected studies, classified as fresh material (F), formulate (FO), and purified (P) compounds.

Type of Tester	Source Organ/Tissue	Extract, Compound, or Class of Compounds ^1^	No. of Papers		References
			Total	With Extract Char.	
F	Seed	HE	1	1	[78]
	Vegetative tissues	Alcohol extract	1	1	[87]
		HE	1	0	[128]
		HE, LE	1	1	[101]
	Unripe fruit	HE, LE, alcohol extract	1	1	[44,97]
	Ripe fruit	HE	6	6	[43,59,61,86,91,95]
		LE	6	4	[67,90,93,98,124,133]
		HE, LE	7	7	[71,75,77,89,104,127,130]
		Lycopene extract	9	7	[62,63,64,94,109,116,123,126,136]
		Anthocyanin extract	2	2	[68,111]
		Digestate	2	2	[35,39]
		Cistine knot-microprotein	1	1	[92]
		Pectic oligosaccharides	1	1	[102]
		Processed (gazpacho, juice, ketchup, paste, passata, soup, sauce, sofrito)	15	14	[38,49,57,59,74,81,82,88,106,108,114,117,119,121,131]
FO		Capsule	6	2	[36,40,54,79,115,118]
		Beadlets (lycopene)	2	2	[80,103]
		Tomato extract	1	1	[37]
		Powder, paste	4	3	[47,70,76,85]
		Seed oil	1	1	[69]
P		Lycopene	23	-	[31,32,33,41,42,46,50,51,53,55,58,60,72,99,100,112,122,125,129,132,134,135,136]
		Tomatine	9	-	[52,56,65,66,83,84,96,105,107]
		Naringenin	3	-	[122,132,137]
		β-Carotene	2	-	[122,132]
		Hydroxytyrosol	2	-	[122,132]
		Curcumin	1	-	[84]
		Tomatidine	1	-	[105]
F and FO		Paste, beadlets	1	1	[45]
		Paste, oleoresins	1	1	[48]
		Juice, capsule	1	1	[34]
F and P		Juice, lycopene	1	0	[113]
		Sauce, lycopene	1	0	[120]
		HE, ketchup, lycopene, vitamins	1	1	[73]
		HE, tomatine	1	1	[97]
		Digestate, tomatine	1	0	[110]

**^1^** HE, hydrophilic extract; LE, lipophilic extract.

**Table 3 foods-10-01905-t003:** “Special” genotypes adopted in research testing tomato nutraceutical properties.

Type	Known Enrichment/Variation	Species/Variety/Genotype Name	Known/Putative Mutation	References
Wild species		*S. habrochaites*, *S. pimpinellifolium*	-	[91]
Breeding lines		*S. pennellii* introgression lines (IL7-3, 12-4)		[75]
	Ascorbic acid	DHO4	-	[95]
Landraces		Italian landraces	-	[78,98,130,133]
		Romanian landraces		[123]
		Central American landraces		[67,130]
		Argentinian landraces		[91]
Mutants	Carotenoids—yellow	Totori Gold, Sugar Yellow, Gold Sugar, V062A, Wapsipinicon Peach, San Marzano giallo, GiàGiù, M-4, M-284; Luracatao yellow	*r* ^2^	[67,71,78,89,91]
	Carotenoids—orange	Gold Minichal, V186A, Jaune Flammee Golden Eye	*B* ^2^	[67,71,124]
		Tangerine line, FG04-169, Olga’s Round Golden Chicken Egg, Golden Green	*t*	[71,85,124]
	Carotenoids—high lycopene	FG99-218	*og^c^ hp-* ^2^	[117]
	Chlorophylls—brown	Brown fruit: Cherokee Purple, Black Kiss, Black tomato ^1,2^ Camone, Black plum	*gf* ^2^	[67,71,78,127,130]
		Zebrino	*gf gs* ^2^	[93]
	Carotenoids—chlorophylls, green	Green fruit: Saeng Green Bichuiball, Saeng Green Chorok	*r gf* ^2^	[71]
	Flavonoids—pink	TI01-0044BABAA, Huachinango	*y* ^2^	[67]
	Anthocyanins—purple	Purple fruit: Sun Black, V118, TI00-0028ADBB, Purple cherry	*Aft atv*	[67,68,77,111,116]
		M1 and M2 (space mutation breeding)		[49]
Genetically modified	Carotenoids	*tlcy-b* overexpressors		[39]

^1^ Used also in hybrid combinations. ^2^ Estimated from the data.

**Table 4 foods-10-01905-t004:** Different target systems and cell line types used in in vitro experiments.

Organism	Type of Line	Target Organ	Cell Lines ^1^	References
Man	Normal	Blood	PBMC (3), macrophages (1), platelets (2), HUVEC (2)	[31,58,59,61,69,73,92]
		Bone marrow	HMSC (1)	[112]
		Breast	MCF-10A (1)	[32]
		Colon mucosa	NCM-460 (1), CCD-18 (1)	[96,104]
		Kidney	HRCE (1), Hek-293 (3)	[75,89,124]
		Liver	Chang (2), WRL-68 (1)	[44,52,71]
		Lung	Hel-299 (2), MRC-5 (1), WI-38 (1)	[44,71,90,107]
		Prostate	PNT-2 (1), RWPE-1 (3)	[52,66,72,84]
		Skin	HSF (1), HaCaT (1), HMEC-1 (1)	[95,98,111]
	Cancer	Blood	HL-60 (3), K-562 (1), U937 (2), THP-1 (3)	[44,51,53,56,73,74,86,113,122]
		Bone	Saos-2 (1)	[131]
		Bone marrow	SH-SY5Y (2)	[105,129]
		Breast	MCF-7 (7), MCF-10A (1), MDA-MB-468 (1)	[32,60,90,93,107,109,116]
		Cervix	HeLa (11)	[44,60,68,71,75,89,90,93,97,116,136]
		Colon	HT29 (10), CaCo-2 (5), SW-480 (3), HCT-116 (2), T-84 (1), LoVo (1), SW-620 (2), SW-48 (1), Colo320D (1)	[33,35,39,41,46,48,49,60,68,96,100,104,110,119,132,136]
		Larynx	HEp-2 (1)	[60]
		Liver	Hep-G2 (5)	[60,71,89,93,116]
		Lung	A549 (4), NCI-H460 (1)	[44,60,71,116,137]
		Prostate	DU-145 (4), LNCaP (4), PC-3 (8), VCaP (1), primary cancer cells (1)	[46,52,60,65,66,72,84,121,124,134]
		Skin	A-375 (1), BEN (1)	[46,78]
		Stomach	AGS (2), YCC-1 (1), -2 (1), -3 (1)	[98,101,102]
Mouse	Normal	Bone marrow	BMDC (1)	[130]
		Embryo	BALB/c3T3 (2), NIH-3T3 (2), SV-T2 (2), 3T3-L1 (1)NIH-3T3 (2)	[75,93,97,102,135]
		Heart	H9C2 (1)	[67]
		Intestine	IEC-18 (1)	[37]
	Cancer	Blood	RBL-2H3 (1)	[43]
		Brain	C6 (1)	[97]
		Colon	CT-26 (1)	[83]
Pig	Normal	Liver	PLP2 (1)	[116]
Other				[87,91,123]

^1^ Numbers in parentheses indicate the number of studies that adopted that specific cell line. Extended names of cell lines are available in the original references and/or at the American Type Culture Collection (ATCC) website (https://atcc.org).

**Table 5 foods-10-01905-t005:** Different target systems used in in vivo experiments.

Organism	Type of Sample	Target	References
Man	Healthy	Adults	[38,48,82,114,118,131,133]
		Sportsmen	[36,57,115]
	Unhealthy	Diabetic patients	[126]
		Asthmatic patients	[34]
		Hypertensic patients	[40]
		Obese females	[88]
		Prostate cancer patient	[54,79,81,117]
Mouse	Healthy	Adults	[37,42,62,63,64,74,99,120,125,127,128]
	Unhealthy	Hypertensic patients	[127]
		Infarcted patients	[55]
		Obese patients	[106,108,135]
		Oxidatively stressed patients	[76,77,85]
		Trombotic patients	[61]
		Blood cancer patients	[56]
		Colon cancer patients	[83]
		Liver cancer patients	[94]
		Prostate cancer patients	[45,47,65,66,70,80,84,103]
		Tongue cancer patients	[50]
Zebrafish			[92]
Drosophila			[113]

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
