# Peer review of "Scientometric and Methodological Analysis of the Recent Literature on the Health-Related Effects of Tomato and Tomato Products"

_foods, 2021, doi:10.3390/foods10081905_

Round 1
Reviewer 1 Report
The authors have addressed the comments.
Author Response
Dear Reviewer, thanks for your positive evaluation of our revision work and resubmitted manuscript.
The Corresponding Author
Reviewer 2 Report
The authors need to follow the following instructions to improve this manuscript.
- The authors should rewrite the abstract based on the best findings and the objectives of the manuscript.
- Line 39 (Mha, Mt): The authors should write elaboration.
- The authors wrote some words as like italic. Why did they write like this? Please clarify or correct. See line 53 (yellow flesh), line 56 (lycopene β-cyclase), line 58 (old-gold crimson), line 60 (tangerine), line 60 (carotenoid isomerase), line 63 (green-flesh), line 73 (colorless fruit epidermis), line 77 (Anthocyanin fruit).
- In the entire manuscript, the authors used 1-2 sentence(s) as a paragraph. This type of writing should avoid. Combine like this paragraph.
- This manuscript should improve based on the foods.
- Line 326 (in vitro and in vivo): The authors may use the italic form here.
- Figure 4: The authors' used a standard bar. Is it deviation/error? How about the replication? The authors should clear the lower bar of the figures.
- The authors should use statistical analysis of all data. Mention it as a superscript.
- The conclusion should precisely write based on the best findings.
- References should check clearly. Check the Journal rules and regulations. Before submission, it is mandatory to check the journal reference writing style.
- The authors used 145 references—too many. I think the authors should use recent and relevant ones, skip the old ones.
Author Response
Dear Reviewer, thanks for your comments. We did our best to address your suggestions. In a few cases we were in doubt on the interpretation of your comment, so we hope to have operated on the suggested direction. In a few cases, we could not add any further improvement and we explained why. We report hereby itemized answers to your comments.
The authors should rewrite the abstract based on the best findings and the objectives of the manuscript.
We have modified the abstract trying to center it as suggested. We believe it is now balanced as it represents an introduction, information on methods, results (main body of the abstract) and conclusions.
- Line 39 (Mha, Mt): The authors should write elaboration.
We guess that the Reviewer means to spell out the measure unit names; we did this correction (New L39)
- The authors wrote some words as like italic. Why did they write like this? Please clarify or correct. See line 53 (yellow flesh), line 56 (lycopene β-cyclase), line 58 (old-gold crimson), line 60 (tangerine), line 60 (carotenoid isomerase), line 63 (green-flesh), line 73 (colorless fruit epidermis), line 77 (Anthocyanin fruit).
Conventionally, gene and mutation names (and the relative abbreviations) are written in italics. Protein names are not italicized. In some journals, extended gene names are not italicized, but we checked that in the MDPI journal “Genes” they do (e.g. https://www.mdpi.com/2073-4425/12/1/59). Therefore we adopted the style used in MDPI journals.
- In the entire manuscript, the authors used 1-2 sentence(s) as a paragraph. This type of writing should avoid. Combine like this paragraph.
We included the small introduction to Materials and methods in Paragraph 2.1. and pooled it with the following paragraph. We think that the paragraph “Statistical analysis” deserves to stay separate as Paragraph 2.2. We included the small introduction to Results in Paragraph 3.1. We pooled the former paragraphs 3.5 and 3.6 in a single paragraph.
- This manuscript should improve based on the foods.
We argue from this comment that more reference to food aspects is requested. To get a higher focus on food aspects, sentences have been added in Introduction (L40-43, L91-94) and Discussion (L412-414, L419-424, L493-497).
- Line 326 (in vitro and in vivo): The authors may use the italic form here.
As explained in our previous responses, we italicized all the occurrences of “in vitro” and “in vivo” as requested by the reviewer, but the English Editing Service at MDPI, that corrected linguistic aspects, returned those terms in normal style. We do not understand why, in this occurrence, the terms should return to italics. We follow the indication on the journal style given by the English Editing Service.
- Figure 4: The authors' used a standard bar. Is it deviation/error?; How about the replication? The authors should clear the lower bar of the figures.
The bars in Fig. 4 represent standard error of the mean; this has been added to the legend (L366). We report means of studies with similar targets, where “n” varies from a minimum of three (“Only targets with at least three inferred values were analyzed and reported” is said in the legend, L365) to a maximum of 21 for colon cancer lines. These values have been now mentioned in the text (L342-343). The dataset is unbalanced, but this has been considered adopting GLM Anova.
We do not understand what is it meant with “The authors should clear the lower bar of the figures.” Does the reviewer refers to the color of the x axis? If so, this has been turned to black. Does he/she refer to delete the error bar in normal prostate lines? In this case, the mean is the result of two values of -6% and +5% (see Table S4) and the bar is the relative correct SEM.
- The authors should use statistical analysis of all data. Mention it as a superscript.
We used GLM and correspondence analysis statistics for the data where we could apply them. Correspondence analysis is based on variable distribution and checks the relativities of such distributions using multivariate chi-squared statistics. It is, to our knowledge, an approach of choice for multivariate analysis of categorical variables. We do not see what other statistical analysis we can apply to counting of countries, journals, cell lines or analysis types.
- The conclusion should precisely write based on the best findings.
The conclusions have been rewritten with the aim to wrap up the most important findings and express the “take-home message” of our work.
- References should check clearly. Check the Journal rules and regulations. Before submission, it is mandatory to check the journal reference writing style.
Since the submission, we have scrupulously followed the journal rules and regulations. In this revision, a number of typos, that inevitably occur within a high number of citations, have been amended, including italics and journal abbreviations.
- The authors used 145 references—too many. I think the authors should use recent and relevant ones, skip the old ones.
We think that 145 is not an excessive number of refs and they are needed in a comprehensive review. For a comparison, a recent review on a parallel topic appeared in Foods 2020 (doi: 10.3390/foods10010045) and cited 250 references (70% more than us). Our bibliography includes the 107 papers analyzed, plus other 38 papers necessary to Introduction and Discussion. In a previous submission of this manuscript, we used 149 refs; one reviewer claimed that they were too much, but at the same time he/she asked for an extension of the search time window and for new search terms (“Bioactive compound”). We addressed all the points asked and kept references in a lower number such as 145; therefore, we believe that it would not positive for our work to further reduce the references.
Round 2
Reviewer 2 Report
The manuscript improved